# Towards Safer T2I Generation by Refining Implicit Prompts

Yijian Lu[1]*   Jiahao Tian[2]†   Yating Liu[3]‡

## 1 Background

With the advancement of Text-to-Image (T2I) technology, high-quality images can now be created effortlessly from arbitrary human-written prompts by models like Stable Diffusion and DALL-E, capturing widespread attention and unprecedented popularity [Saharia et al., 2022, Takagi and Nishimoto, 2023, Betker et al., 2023]. However, this flourishing T2I community has also led to increasing privacy concerns, particularly regarding celebrity privacy [**?**]. The unauthorized generation of celebrity images can lead to the spread of misinformation and the destruction of reputation.

To tackle this problem, many T2I models are equipped with an addition safety checker to filter out user prompts containing celebrity names [Saharia et al., 2022, Lee et al., 2024]. Such filters offer a simple yet effective approach to reduce the privacy threats. However, implicit prompts, which obviously suggest a celebrity figure without directly containing the name, pose a more subtle risk to privacy. For instance, instead of explicitly referring to *"Taylor Swift"*, implicit prompts such as *"American female singer known for her country-pop hits and songwriter, like 'Love Story'"* may also generate a precise portrait of *"Taylor Swift"* [**?**]. These implicit description poses latent threat to celebrities' privacy and reputation. Therefore, a corresponding strategy is urgently required to handle implicit prompts.

In this project, we firstly design a method to enhance the defense ability of T2I models to implicit prompts. Moreover, to go beyond than simply rejecting implicit prompts, we propose to conduct refinement on user prompts that preserve the most information while not lead to final images of celebrities. All above-mentioned process are free from modifying the T2I models to preserve the original capabilities and ensure the quality of generated images intact. We hope this project could make certain contribution to the responsible application of T2I technology and foster advancements in ethical AI practices.

## 2 Definition

The primary goal of this research is to develop a machine learning framework that identifies and mitigates the generation of images associated with implicit prompts involving celebrity identities. Specifically, our goal is first to identify celebrity-targeted images $U$ generated by original prompts $P$, and then to produce safe images $I$ without any celebrity implication, where

- $P$: Initial user prompts input to a T2I model;
- $U$: Unwanted output, i.e., generated image that implicitly refers to a specific celebrity based on initial prompts;
- $I$: Generated image from the T2I model based on refined prompts.

*[1]Department of Computer Science and Technology, `luyj24@mails.tsinghua.edu.cn`

†[2]Institute for Interdisciplinary Information Sciences, `tianjh24@mails.tsinghua.edu.cn`

‡[3]Department of Automation, `liuyt24@mails.tsinghua.edu.cn`

38th Conference on Neural Information Processing Systems (NeurIPS 2024).

# 3 Related Work

**Text-to-image Generation.** Text-to-image (T2I) generation is the process of creating images from textual descriptions using machine learning models. Early T2I generation techniques relied on generative adversarial networks (GANs)[Creswell et al., 2018, Tao et al., 2022], which faced challenges with scaling effectively and mode collapse. Currently, diffusion models are the most commonly used technique for T2I generation, known for their stable training process and ability to produce high-quality images[Ho et al., 2020, Rombach et al., 2022].

**Safety of T2I Models.** As the capabilities of this technology continue to advance, some researchers have begun to consider its implications for social security[Rando et al., 2022, Schramowski et al., 2023]. For example, T2I models can be prompted to generate images containing celebrity likenesses, which could potentially violate celebrities' privacy and rights[Yang et al.]. To achieve this, people might use either explicit or implicit prompts - while explicit prompts can be easily detected and filtered, implicit prompts can bypass these filters and raise subsequent concerns. Currently, there are two main strategies for generating images without unwanted contents using T2I generation.

The first is removing unwanted images from the training set, such as excluding all celebrity images[Nichol et al., 2021]. However, this method requires model retraining, which not only increases costs but may also lead to a decline in the quality of generated images[Zhang et al., 2024b]. The second is a post-hoc strategy, such as implementing blacklists after model completion to prohibit unsafe image outputs[Markov et al., 2023]. Other approaches include using expensive fine-tuning to eliminate inappropriate content during model training, or identifying inappropriate content in input prompts before the diffusion process begins[Zhang et al., 2024a, Park et al., 2024]. Now, research on celebrity privacy and implicit prompts is limited. One experiment revealed that existing T2I models are more likely to successfully generate images of higher-profile celebrities, posing serious risks of misinformation spread and damage to celebrities' personal reputations. The researchers proposed an implicit prompt benchmark at the input stage to adjust prompt reformulation models, suggesting this could effectively reduce the generation of image that violates celebrity privacy[Yang et al.].

# 4 Proposed Method

Our method consists of two steps, prompt inspection and prompt refinement. The first one is to detect the input prompt, which determines whether it clearly refers to a celebrity, leading to the T2I model generating outputs that could compromise the privacy of that celebrity. The second step, prompt refinement, involves modifying the implicit prompt in a way that, while ensuring the output does not contain images of the celebrity, maximizes the retention of the original prompt's information so that the output image aligns with the requirements of the prompt. The following paragraphs provide explanations about the motivations and details of our proposed method for each step.

To identify the implicit prompt, we propose to design a post-hoc detector based on face recognition models [Wang and Deng, 2021]. Specifically, the T2I model will generate the corresponding images using the original prompt without being influenced by any external disturbances. Then using the face recognition model, if the image is recognized as a public figure with high confidence, the prompt will be labelled as an implicit prompt. Only positively-labelled prompts will go through the next refinement step. The reasons why we adopt a post-hoc detector are two-fold: 1) to make sure the positively labelled prompts are **indeed** implicit, as images resulting from some seemingly-implicit prompts are safe; 2) to facilitate the prompt refinement process where features extracted from the generated image could be used to complete/replace the implicit prompts.

The second process is to modify the implicit prompt $P$ to generate safe yet relevant images. To do that, we propose a collect-then-select approach. Intuitively, a simple way to modify the prompt is to exclude all descriptive words that lead to a particular celebrity from the prompt. However, the given implicit prompts are likely to contain mostly such information so that after the removal is conducted, too few words are left in the prompt. Therefore, we propose to first **collect** features $X$ about the implied celebrity from the previously-generated image using a Image-to-Text model, then perform the prompt reconstruction by gradually removing prompt components. The **select** phrase can be achieved by either utilizing a LLM [Zhao et al., 2023] api or fine-tuning a Seq2seq model [Raffel et al., 2020]. The aim here is to maximize the size of the remaining feature set $X'$ while making sure the resulting image is celebrity-safe.

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
