# OpenReview forum: "[Proposal-ML] Towards Safer T2I Generation by Refining Implicit Prompts"
_tsinghua.edu.cn/THU/2024/Fall/AML — THU 2024 Fall AML Submission_

### Official Review · ~Chua_Shei_Pern1 · 2024-11-06
**Good**

**Rating:** 9
**Confidence:** 4

**Review:**

The proposal presents a timely and well-structured approach to address privacy concerns in text-to-image (T2I) models, particularly focusing on implicit prompts that could lead to unwanted celebrity likenesses. While the methodology is sound, adding more specifics on evaluation metrics could further clarify the project's impact and feasibility.

---

### Official Review · ~Un_Lok_Chen1 · 2024-11-08
**A Novel Pipeline on Tackling the Implicit Prompt Problem in T2I Generation Task**

**Rating:** 8
**Confidence:** 4

**Review:**

Summary:

This project proposal aims at tackling the implicit prompt problem in text-to-image task that poses risk in celebrities’ reputation. Their proposed method consists of two stages: prompt inspection and prompt refinement. In the prompt inspection stage, the authors propose to use an image recognition model to detect if the generated image contains celebrity faces. If so, they will apply a collect-and-search strategy to refine the text prompt while retaining most original image features.

Pros:

1. The research problem and motivation is well stated and considered critical. The specific focus on implicit prompts about a celebrity is carefully selected and appropriate.

2. The overall framework of the proposed method is reasonable and relatively detailed, and the authors provide much intuition into why they construct such  a pipeline.

Cons:

A. Major issues:

1) Consider adding details on how the experiments will be implemented specifically (e.g. datasets, training method, evaluation schemes).

B. Minor issues:

1) There are missing in-text citations in several parts of the Background section (indicated by the ? mark).

2) The Definition section is a bit redundant, consider highlighting the primary goal of this project more succinctly.

3) Are there existing methods that also adopt similar post-hoc methodology in the literature? How is the proposed method different from/connected with other related works?

---

### Official Review · ~Joydeep_Chandra2 · 2024-11-08
**Innovative Strategy for Implicit Prompt Detection in T2I Models, Yet Needs Clearer Execution and Impact Assessment**

**Rating:** 7
**Confidence:** 3

**Review:**

Innovative methodology with clear steps outlined. The use of face recognition and Image-to-Text models for implicit prompt detection is well-justified. But, It could provide more specific real-world examples or case studies illustrating the misuse of implicit prompts for stronger context. Some places [?] is used which i am unaware what is the significance in the paper. The methodology also lacks detailed implementation plans, such as specific metrics for evaluating the effectiveness of prompt refinement.

---

### Official Review · ~Diego_Cerretti1 · 2024-11-10
**Well-structured and relevant**

**Rating:** 8
**Confidence:** 4

**Review:**

The paper aims to tackle an emerging issue in AI: the improvement of the safety and ethical standards of text-to-image models. The approach proposed by the authors is well-structured and clear, with the post-hoc detector being a well-justified technique to identify implicit prompts. The greatest issue with the proposed approach is the lack of specified performance metrics or benchmarks to assess the success of the proposed solution. It is unclear how the experiments will be carried out and evaluated.

---

### Official Review · ~Rim_El_Filali1 · 2024-11-11
**Good Approach to Enhancing T2I Safety via Implicit Prompt Refinement but Needs Structured Evaluation Plan**

**Rating:** 7
**Confidence:** 4

**Review:**

This proposal tackles the timely issue of privacy concerns in T2I generation, specifically focusing on celebrity image generation from implicit prompts. The paper proposes a method to detect and refine these prompts to prevent unintentional celebrity image generation while preserving the information in the original prompts.

Pros:
- The research addresses an important aspect of T2I technology, emphasizing the ethical and privacy concerns around celebrity image generation.

Cons:
- The proposal does not detail a structured plan for evaluating the effectiveness of the refined prompts in both protecting privacy and maintaining prompt intent.
- Missing citations in some parts of the Background section.

---

### Official Review · ~Fabian_Pawelczyk1 · 2024-11-11
**Strong Proposal**

**Rating:** 8
**Confidence:** 4

**Review:**

**Decision: Clear Accept**

This proposal is clear and directly addresses a key issue in Text-to-Image (T2I) generation: managing implicit prompts that can create recognizable celebrity images. The approach is timely and necessary because, while current safety filters block prompts that directly name celebrities, they do not address indirect descriptions that still pose privacy risks. The project’s focus on refining prompts to remove these indirect identifiers without changing the T2I model itself is both practical and realistic.

The proposed two-step method, a face recognition check after image generation and a "collect-then-select" approach to prompt refinement, effectively tackles the problem without reducing model quality. This solution respects user intentions and supports responsible AI use by adjusting rather than blocking prompts, making it suitable for a classroom setting and offering a smart approach without requiring major model modifications.

It would be helpful to include more details on which features are prioritized during prompt refinement to make the process clearer and easier to replicate. Additionally, there are some minor issues in formatting and citations, which could be improved for better readability and precision.

Overall, this is an stong proposal with only a few minor areas for improvement, making it a strong candidate for approval.

---

### Official Review · ~Michael_Hua_Wang1 · 2024-11-11

**Rating:** 9
**Confidence:** 4

**Review:**

The use of generative AI remains contentious to this day, and one particular sticking point has been the use of "deepfakes" to create imagery that can have harmful effects on a person's reputation and livelihood. Celebrities are especially vulnerable due to their public visibility.

This proposal seeks to take on an important issue with respect to implementing the guardrails required to ensure safe and ethical use of generative AI tools.

However, I think the proposal could afford to spend a bit more time describing the method used for prompt reconstruction, as that appears to be one of the most significant factors in how successful the authors' approach will be.

---

### Official Review · ~Zhaoxi_Li2 · 2024-11-12
**Proposal Review: Towards Safer Text-to-Image (T2I) Generation by Refining Implicit Prompts**

**Rating:** 9
**Confidence:** 4

**Review:**

This proposal addresses a critical issue in text-to-image (T2I) generation by focusing on the identification and refinement of implicit prompts that may inadvertently generate images of celebrities, posing privacy concerns. The proposed method enhances the safety of T2I models by employing a two-step process: an initial prompt inspection to detect implicit celebrity references, followed by prompt refinement that maintains content integrity without producing celebrity likenesses. The authors’ approach is well-considered, utilizing a post-hoc detection mechanism based on face recognition and a prompt refinement step that maximizes alignment with the original prompt's intent. This strategy is thoughtfully designed to preserve the T2I model's original capabilities while enhancing safety, contributing to ethical AI development. However, further details on the expected efficacy of prompt refinements across a range of implicit descriptions and the generalization of the detection model across diverse T2I models would strengthen the proposal. Overall, this work has strong potential to advance responsible T2I model usage and address a pertinent aspect of AI ethics in image generation.

---

### Official Review · ~Qihang_Cen1 · 2024-11-12
**Clear Problem Definition with Practical Relevance**

**Rating:** 8
**Confidence:** 4

**Review:**

The paper clearly articulates the privacy risks associated with implicit prompts in T2I models. It is meaningful and relevant to focus on protecting celebrity privacy from implicit prompts. The proposed two-step method and “collect-then-select” strategy are logically sound and practical. The proposal also references a wide range of studies that support its approach, demonstrating a strong theoretical basis for the proposed solution. Howerer, in this proposal, maybe add experiments design details would strengthen the paper's credibility, while disucssion on impact on image quality might also important.

---

### Official Review · ~Kaiyuan_Zhang6 · 2024-11-12
**Well done proposal**

**Rating:** 10
**Confidence:** 5

**Review:**

The work aims to solve the safe problem in T2I generation area. The methods divided as two part: identify the implicit prompt, and modify it. Each part of the methods is clearly discussed, with relative enough techinique details and future plans. Besides, background and related work is quite well written. One single concern that not mentioned is how to evaluate the final generation result, which should contain the quality of the refined prompt and image, as well as the information consistency.

---

### Official Review · ~Zhixuan_Pan1 · 2024-11-12

**Rating:** 8
**Confidence:** 4

**Review:**

This project proposes a framework to make Text-to-Image (T2I) generation safer by detecting and refining implicit prompts that could lead to celebrity likenesses. The goal is to modify prompts in a way that retains the original prompt's intent while preventing unauthorized celebrity representations.

Pros:

1.Addresses a novel privacy concern by targeting implicit prompts in T2I, contributing to ethical AI practices.
The two-step approach (detection and refinement) maintains T2I model integrity while enhancing output safety.

Cons:

1.If only prompt engineering is used, ensuring safety on a larger scale may require lengthy prompts, which could negatively impact the model's performance. Fine-tuning may be required.

---

### Official Review · ~ChenJian1 · 2024-11-12
**Brief review**

**Rating:** 8
**Confidence:** 3

**Review:**

The proposal aims to enhance the safety of Text-to-Image (T2I) generation technology by improving the handling of implicit prompts. The authors propose a machine learning framework to identify and mitigate the generation of images associated with implicit prompts involving celebrity identities. The framework includes two main steps: prompt inspection and prompt refinement, ensuring that the output images do not contain images of celebrities while retaining as much information from the original prompt as possible.

### Strengths:
Innovative Approach: The proposal addresses a significant issue in the current AI ethics and privacy protection domain by introducing a novel method to handle implicit prompts in T2I models.

### Weaknesses:
Experimental Validation: The proposal lacks experimental results to support the effectiveness of the proposed method, which may affect the credibility of the approach.